# Estimating and abstracting the 3D structure of feline bones using neural networks on X-ray (2D) images

Jana Čavojská [1✉], Julian Petrasch[1,2], Denny Mattern [3], Nicolas Jens Lehmann [1], Agnès Voisard[1] & Peter Böttcher[4]

Computing 3D bone models using traditional Computed Tomography (CT) requires a high-radiation dose, cost and time. We present a fully automated, domain-agnostic method for estimating the 3D structure of a bone from a pair of 2D X-ray images. Our triplet loss-trained neural network extracts a 128-dimensional embedding of the 2D X-ray images. A classifier then finds the most closely matching 3D bone shape from a predefined set of shapes. Our predictions have an average root mean square (RMS) distance of 1.08 mm between the predicted and true shapes, making our approach more accurate than the average achieved by eight other examined 3D bone reconstruction approaches. Each embedding extracted from a 2D bone image is optimized to uniquely identify the 3D bone CT from which the 2D image originated and can serve as a kind of fingerprint of each bone; possible applications include faster, image content-based bone database searches for forensic purposes.

[1] Freie Universität Berlin, Institute of Computer Science, Berlin 14195, Germany. [2] Isar Aerospace Technologies GmbH, Ottobrunn 85521, Germany.
[3] Fraunhofer FOKUS, Data Analytics Center, Berlin 10589, Germany. [4] Freie Universität Berlin, Clinic for Small Animals, Berlin 14163, Germany.
✉email: jana.cavojska@fu-berlin.de

Traditionally, computation of three-dimensional (3D) bone models is based on computed tomography (CT) scans, resulting in high-radiation dose, cost, and time consumption. In the veterinary field, CT image acquisition also involves full anesthesia of the animal during scanning, making this approach an invasive and highly expensive procedure. Generating 3D models directly from two-dimensional (2D) images can be a useful alternative[1]. Currently, multiple approaches for estimating the 3D structure of bones from their 2D X-ray images exist[1–8], which help reduce the cost and the radiation-related health risks for the patient, as well as the necessity of anesthesia in most of the animals. There are even a few fully automated approaches for bone shape estimation[9,10]. However, one such approach requires previous knowledge about the bone geometry in order to identify bone boundaries in the input image[9], and the other requires five X-ray images taken from different angles[10].

In recent years, many deep-learning-based approaches to 3D object reconstruction from 2D images emerged[11–20].

The approach by Henzler et al.[12] uses a neural network to generate a 3D shape from a single bone radiograph, with the goal to recover 3D data for databases of fossils where only 2D data are available. The main drawback of this method which operates in the absence of any priors is that it can generate implausible 3D shapes, such as skulls that do not optically resemble skulls.

Other methods exist that attempt to reconstruct the 3D shape of an object from a single image[11], using, for example, segmentation masks and key points to build category-specific shape models[13] or using surface normal prediction[14] or mesh reconstruction while exploiting shading and lightning information[15]. However, in the absence of either multiple viewpoints or previous knowledge about object geometry, no accurate reconstruction of the occluded parts of the object can be guaranteed.

The recurrent 3D-R2N2 network by Choy et al.[16] learns a mapping from observations to the underlying 3D shapes of objects from a large collection of training data. The network first generates an embedding of the 2D image and then reconstructs the object in the form of a 3D occupancy grid based on this embedding. A single input image is sufficient for the 3D reconstruction; however, if multiple images of the same object from different views are available, they are used to refine the initially estimated 3D shape. This method suffers from the problem that when a set of 2D input images is fed into the network in a different order, it produces different reconstruction results[17].

Xie et al.[17] address this and other problems using their Pix2Vox framework for single-view and multi-view 3D reconstruction. Their encoder–decoder first generates a coarse 3D volume from each input image. Then, a context-aware fusion module adaptively selects high-quality reconstructions for each part (e.g., table legs) from different coarse 3D volumes to obtain a fused 3D volume. Finally, a refiner refines the fused 3D volume to generate the final 3D output.

Both 3D-R2N2 and Pix2Vox learn a discrete embedding space from which the 3D shapes are reconstructed. Others[18,19] propose continuous embedding spaces by incorporating the capabilities of a variational autoencoder (VAE) into their pipeline.

Wu et al.[18] propose a 3D-VAE-GAN architecture where a GAN (generative adversarial network) is trained to generate the 3D shape from a latent space, and a VAE is used to ensure the latent space is continuous.

Liu et al.[19] work with a hierarchical continuous latent space, meaning that instead of using a single embedding vector as the intermediate representation, they generate a more complex internal variable structure consisting of one global latent variable layer hardwired to a set of local latent variable layers, each representing one level of feature abstraction. This more complex structure aims to improve the quality of the GAN reconstruction and to prevent the blurriness of the reconstructed images.

We present a fully automated neural network-based method for estimating the 3D structure of a bone from a pair of orthogonal 2D X-ray images. Like many deep-learning-based methods in general, our method is completely domain-agnostic, which means that all that is needed for teaching the network to work with completely different bones, such as human tibias instead of cat femurs, is a set of 3D CT images of such new bones. Our method is based on assigning the most closely matching 3D shape of a bone to the 2D input image of that bone by selecting the shape match from a pre-existing set of 3D shapes. We treat the search for an optimal 3D shape as a classification problem. Each class corresponds to one specific 3D shape (i.e. one specific bone) and is presented to the neural network during training as a set of 2D images generated from the 3D shape. To build a classifier that solves this problem, we trained a convolutional neural network (CNN) using the triplet loss[21–29] method and taught it to differentiate between femurs of different cats based on artificial 2D X-ray images generated from the 3D femur CT scans. Once fully trained, the network was able to generate for each input image a low-dimensional representation of its content, a so-called $d$-dimensional image embedding. We then trained a $k$-nearest neighbor (kNN) classifier on these embeddings. To predict the shape of a new bone, the kNN classifier assigned the 2D input image embedding of that bone to its closest bone match from the training set. Since the 3D shapes of the bone images from the training set are known, this assignment results in identifying the 3D shape that most closely matches the bone in the input image, as determined by the properties (features) extracted by the neural network.

How well these properties, which were considered relevant by the network, correlate with the bones' actual 3D shapes was evaluated by computing the root mean square (RMS) and Hausdorff distances[30,31] between sample bones and the ground truth (the actual 3D shape of the sample bone). Neither the network nor the kNN classifier was trained on the sample bones. To put these evaluation results into context, the measured distances were also compared with the distances obtained using statistical shape models (SSMs) for 3D shape reconstruction of the same bones. The evaluation results were also compared with distances achieved on similar tasks in related literature.

Training a neural network using triplet loss results in the network's ability to generate embeddings (vectors) that can be used to uniquely identify the specific bone the input image depicts. These embeddings are easily separable by their Euclidean distance, even in case of bones that were never presented to the network in any way during training. Embeddings of artificial X-ray images generated from the same CT scan build tight groups in the Euclidean space, while embeddings from different CTs and hence different bones are distinctly further apart, which can be used for data compression purposes and for searching bone databases.

The existing deep-learning based methods that generate an embedding of the 2D image as an intermediate representation[16–19] optimize their embeddings to hold 3D shape information, while our bone embeddings are optimized to uniquely identify highly similar 3D objects, while still making 3D shape inference possible.

Jointly training one of the networks that relies on embeddings with our network can therefore result in embeddings that combine these capabilities and would be a very interesting future application. It remains to be seen whether the pairs of orthogonal images in our bone dataset contain sufficient information for such a shape reconstruction and whether the generated 3D shapes are suitable for clinical purposes.

We have found that using a pre-trained, generic neural network classifier, we already achieve a 100% classification accuracy on our bone dataset. We achieve the same result using a Triplet network as a feature extractor followed by a $k$-nearest neighbor

classifier. We show that the extracted features (image embeddings) can be used not only for classification and hence shape estimation, but also for pairwise comparisons between X-ray images, based on the differences between fingerprints of different bones. An examination of the statistical properties of the embedding space shows that after triplet training is complete, some of the embedding dimensions remain redundant, opening up possibilities for further compression. We finally show the results of our abstraction-based classification approach to shape estimation, first qualitatively by showing alignments between our predictions and the true 3D shapes and then quantitatively: We first compare our average RMS distance of 1.08 mm between predicted and true shapes with the distances achieved by different methods found in other literature (1.32 mm, on average), and then we use a traditional SSM on our bone data (3D CT scans as well as separately made, natural 2D X-ray images of the same bones). In two-thirds of the examined cases, our Triplet network/ kNN classifier finds better shape matches than the SSM approach on the same data (measured by the RMS distance between predicted and true shapes). We further found that when deformities are introduced into the 2D bone images, depending on the extent of the deformity, a deformed bone can only still be successfully matched with 57–92% of the images of the same bone without said deformity. The shape prediction accuracy deteriorates by approx. 25% for the deformed bones. However, not a single instance of a deformed bone being matched with the wrong healthy bone was observed—the deformed bone was only matched to its healthy counterpart from before the deformation.

## Results

**Dataset properties**. Our dataset consisted of 29 3D CT scans of 29 femurs of different cats in the DICOM format. These scans were provided by the Clinic for Small Animals, Freie Universität Berlin. We chose to work with a dataset of 29 bones because it provides a good compromise between roughly representing the variety of feline femurs that are commonly encountered in clinics (consisting of specimens that differ in their lengths, widths, and other shape variations) and being small enough that it is easy to generate, should the need arise to expand our approach to other bones than femurs, with a comparable accuracy. For the purpose of additional validation, three new CT scans were added, each paired with the corresponding two natural X-ray images of the same bone as in the CT scan; the first natural X-ray image showed the bone from the anteroposterior and the second from the mediolateral view. Using the software *MeVisLab*, we generated 900 artificial 2D X-ray images for each of the 29 CT scans by varying the viewing angle and radiation exposure, thus simulating the variability of conditions during clinical X-ray image acquisition[32,33]. These 26,100 images served as the dataset to train and evaluate the neural network on. For evaluation purposes, the 3D shape in the STL format was extracted from each 3D CT scan so that the difference between a predicted shape and the true shape could be measured. Finally, we also evaluated the accuracy of our shape estimation and feature abstraction on 2D images of artificially deformed bones. The different data types are shown in Fig. 1.

**Neural network-based classification results encourage a feature extraction approach**. We first tested neural networks' ability to extract meaningful features from 2D X-ray images of bones by performing transfer learning[34] on the dataset of 26,100 images of cat femurs artificially generated from 3D CT scans. Different network architectures pre-trained on the ImageNet[35] benchmark dataset were used. The last layer of these networks, the dataset-specific softmax layer[36], was removed and replaced by a softmax layer specific to our bone image dataset. The network, modified

this way, was then further trained on our bone image dataset. It made no difference for the achieved accuracy whether only the new softmax layer was trained or the earlier layers were fine-tuned, as well. Implementations using different framework and neural network combinations were tested: (i) Keras with the ResNet-50 (ref. [37]) network and a validation accuracy of 86.13%; (ii) Keras with the VGG-16 (ref. [38]) network and a validation accuracy of 63.13%; (iii) TensorFlow with the Inception-ResNet-V2 network and a validation accuracy of 85.80%; (iv) TensorFlow Hub with the ResNet-50 or Inception-V3 networks and a validation accuracy of 100.00%.

Neither different optimizers (Stochastic Gradient Descent, Adam) nor different learning rates (0.01, 0.0001) made a difference in accuracy greater than 1%.

The transfer learning experiments showed that neural networks have the ability to classify bones based on their artificial X-rays with a high accuracy, 100% in case of the TensorFlow Hub implementation. A neural network classifier trained using transfer learning could be sufficient to estimate the 3D structure of bones: This classifier can be used to predict the class, i.e. the specific 3D shape most closely matching the bone in the 2D input image. The reason why we chose the triplet loss method over transfer learning was that we were looking for a way to extract features from the input data that could serve purposes other than simple classification. We decided to use triplet loss over similar distance metric learning approaches such as contrastive[39] or magnet[40] loss because recent publications[22,23] have shown how successful it is in extracting visual features from large datasets, even for tasks such as face recognition, as demonstrated by the 99.63% accuracy the FaceNet network achieves on the widely used Labeled Faces in the Wild (LFW) dataset and the 95.12% accuracy on the YouTube Faces Database[22]. Some publications argue for using softmax combined with metric learning[23], or for using magnet loss[40] over triplet loss, while others defend triplet loss against these approaches[23,41]. A drawback of the representations learned through softmax is that they attain limited intra-class compactness and inter-class separation when compared to triplet embeddings[41]. In addition, the publication defending magnet over triplet loss verifies their claims on datasets of no more than 120 classes, whereas Schroff et al.[22] use their FaceNet Triplet network on 1595 classes in case of the YouTube Faces Database and 5749 classes in case of the LFW database, achieving the above-mentioned high accuracies. This was a strong argument for us to use triplet loss, because it indicates the robustness of this loss function at scale (for thousands of classes) if our approach was to be expanded to use a much larger amount of different bones in the future. One possible use case can then be to augment large forensic bone databases with embedding data, so that each bone could be searched for using only its image embedding.

**Attention map visualization shows no dataset deficiencies**. In order to verify whether the networks used for transfer learning were capable of extracting the right features, we applied the grad-CAM (Gradient-weighted Class Activation Mapping)[42] visualization to images from the 29 classes. The grad-CAM visualization approach identifies the regions of the input image which are most relevant for predicting a certain class. It does so by analyzing the gradients flowing into the last convolutional layer of a network. Specifically, in order to find the regions of the image most relevant to a specific class, Selvaraju et al. compute the gradient of the score for that class with respect to the feature maps of the last convolutional layer. The last convolutional layer is chosen for this purpose because it has been shown that deeper network layers capture higher-level visual constructs (such as whole objects, as opposed to simple edges or textures). And unlike fully connected layers, convolutional layers of a

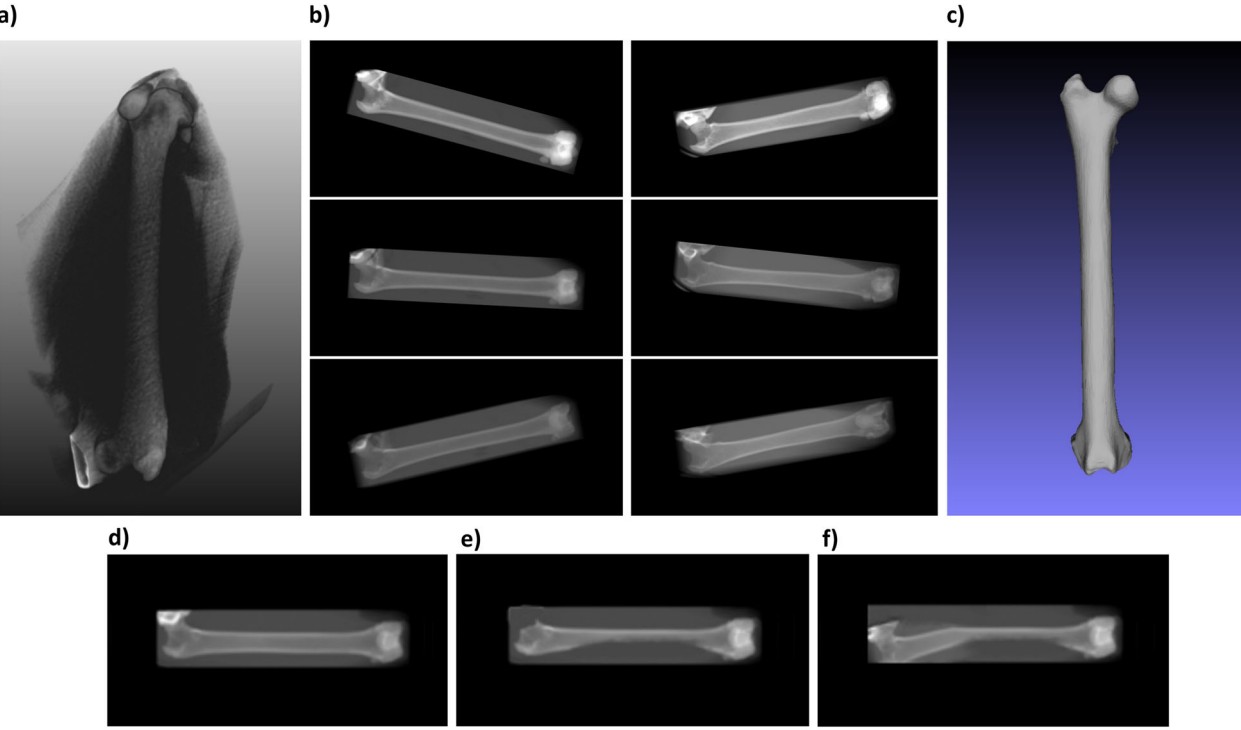

**Fig. 1 Dataset. a** 3D CT DICOM file. **b** Examples of X-ray images artificially generated from 3D CT DICOM data. Images in the left column of **b** were generated from the same bone. Images in the right column of **b** were generated from a second bone. **c** Bone mesh (surface model) extracted from the 3D CT DICOM file. **d** Artificial X-ray image of a healthy bone. **e** Artificial X-ray image of a deformed bone. **f** Artificial X-ray image of a strongly deformed bone.

network retain the spatial information from the input image which makes it possible for grad-CAM to localize the most relevant complex features in the input image[42].

Figure 2 shows the results of the grad-CAM visualization applied to the VGG-16 network after transfer learning. Each blue square represents one of the 29 bones, and the bright red overlays highlight the regions of an image which were most relevant for its correct classification. On almost all the images (except for the penultimate one), the bright red region coincides with a part of the bone. In most images, both bones are being taken into consideration. This visualization has not shown any obvious defects in the dataset, such as the network identifying artifacts (such as the crop boxes around the bones) or any other unexpected anomalies which could be easily remedied. Even in the cases where a big portion of the relevant regions lies outside of both bones, the most relevant region still overlaps with the bones. This is a coarse indication that the features extracted by the network do not have any obvious defects and augmenting the dataset further is not necessary.

**Using feature extraction with triplet loss on the bone dataset.** In order to be able to use the features extracted from the 2D X-ray images for a variety of purposes, not only classification, we separated the feature extraction step from the classification step in a way also used in face recognition and person re-identification publications[22,23].

We modified the network architecture ResNet-50 (ref. [37]) and later VGG-16 (ref. [38]) and incorporated these architectures, in consecutive experiments, into a Triplet network architecture which was then trained with the triplet loss method. When performing transfer learning, the neural network is trained by iteratively being shown examples of each class, computing the error (difference between predicted class and true class) and modifying the network parameters to decrease this error. The Triplet network was instead trained by being iteratively shown groups of three images, so-called triplets: the first image of a triplet, so-called *anchor* ($x_i^a$), was selected randomly. The second image, *positive* ($x_i^p$), was selected from the same class as anchor. The third image, *negative* ($x_i^n$), was selected from a different class than anchor. The error was computed as follows: For each of the triplet images, its low-dimensional representation ($d$-dimensional embedding) was computed. The embeddings were computed by the *base network* part of the Triplet network, consisting of either ResNet-50 or VGG-16, followed by dimension reduction layers. Each embedding was normalized to have unit length, i.e. the embeddings were forced to live on the surface of a $d$-dimensional hypersphere. The Euclidean distances between the embeddings of anchor and positive, as well as between the embeddings of anchor and negative, were computed. Like in the work of Schroff et al.[22], the loss $L$ to be minimized, i.e. the "triplet loss", was then computed as

$$L = \sum_i^N \max\left(0, \, \left\|f(x_i^a) - f(x_i^p)\right\|_2^2 - \left\|f(x_i^a) - f(x_i^n)\right\|_2^2 + \text{margin}\right),$$

(1)

where

- $f(x_i^a)$ is the network's representation ($d$-dimensional embedding) of the anchor image. The embedding dimension, $d$, is selected prior to training.
- $f(x_i^p)$ is the embedding of the positive image.
- $f(x_i^n)$ is the embedding of the negative image.
- $\left\|f(x_i^a) - f(x_i^p)\right\|_2^2$ is the squared Euclidean distance between the embeddings of the anchor and the positive image.
- $\left\|f(x_i^a) - f(x_i^n)\right\|_2^2$ is the squared Euclidean distance between the embeddings of the anchor and the negative image.
- margin is the minimum Euclidean distance between embeddings of different classes enforced by the triplet loss function during training.

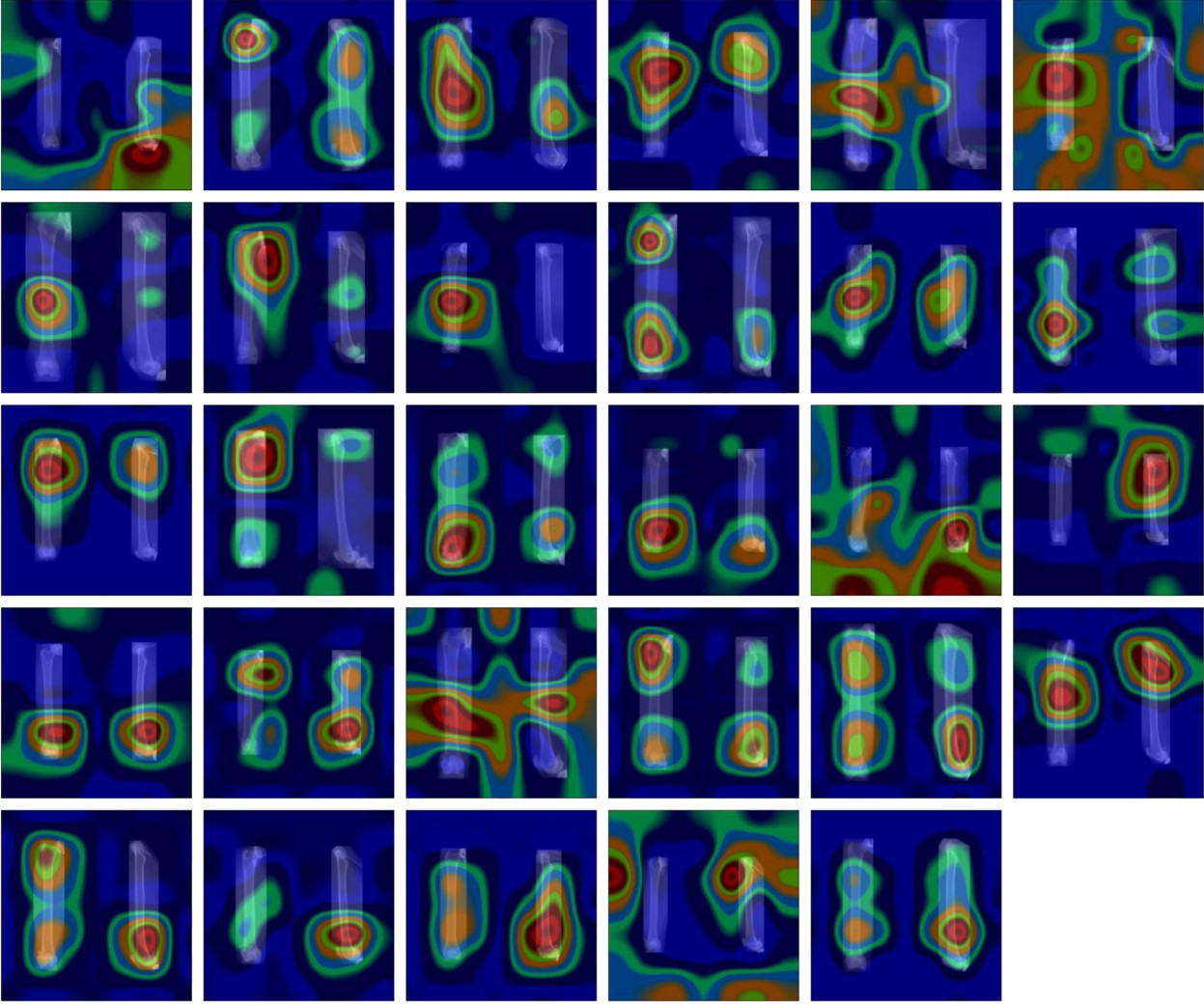

**Fig. 2 Attention maps overlaid over the bone images of the 2D image dataset.** Each of the 29 images shows one of the cat femurs from the anteroposterior and mediolateral view next to each other, overlaid by a heatmap which highlights in bright red the regions of the image that were most relevant for its correct classification using the VGG-16 network.

This loss function was designed to minimize the Euclidean distance between the anchor embedding and the positive embedding while maximizing the distance between the anchor embedding and the negative embedding. The loss function also enforces a minimum distance of margin between the embeddings of different classes. It did not make any difference in accuracy whether or nor the Euclidean distance was squared.

The triplet accuracy function was then defined as the percentage of triplets for which the following condition was met:

$$\left\| f\left(x_i^a\right) - f\left(x_i^p\right) \right\|_2^2 + \mathrm{margin} < \left\| f\left(x_i^a\right) - f\left(x_i^n\right) \right\|_2^2. \quad (2)$$

The Triplet network accuracy was computed as the percentage of triplets for which the Euclidean distance between same-class image embeddings was smaller, by margin, than the Euclidean distance between different-class embeddings.

We conducted a series of neural network training experiments to examine the influence of different hyperparameters and other factors on the resulting Triplet accuracy. Each experiment was repeated at least three times to ensure that the resulting accuracy stayed within a 0.2% interval.

The optimal value for the hyperparameter margin was experimentally determined to be 0.1 for the bone dataset. The

value 1.0 lead to an accuracy drop of 9.6%, making it the the parameter which influenced the accuracy most.

In related work[22,43], $d$ is set to 128. We tested the embedding dimensions 128, 64, 32, and 16. An accuracy drop (approx. 1%) was only noted with 16-dimensional embeddings. After these tests, we conducted all experiments with $d = 128$.

The validation accuracy of the Triplet network achieved 99.9% when the following options were used: a margin value of 0.1, using 900 images per class, X-ray images showing bone from anteroposterior next to a mediolateral view, 128-dimensional embeddings, using the VGG-16 network rather than ResNet-50, and using X-ray images with bones scaled relatively to their real-world proportions. The choice of the hyperparameter margin had by far the biggest impact. The second most important option after margin was a sufficient number of dataset images; when the number was decreased to 100 images per class, a validation accuracy drop of 3.5% was observed. Using images with the combined anteroposterior and mediolateral view increased accuracy by 2.0% as opposed to only using anteroposterior images. Using variable learning rate as opposed to a constant one for all layers improved the accuracy by 1.0%. Replacing ResNet by VGG as part of the base network within the Triplet network improved the accuracy by 0.8%. Scaling bones

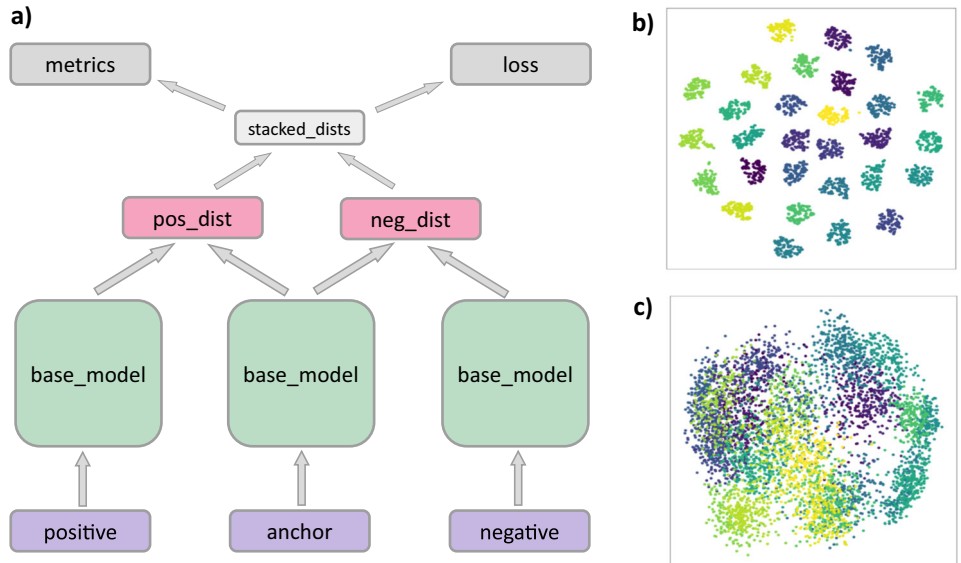

**Fig. 3 Triplet network architecture and visualizations of the bone embedding distances it generates. a** Architecture of the Triplet network: three identical copies of the base network (VGG-16 or ResNet-50 followed by dimension-reducing, embedding-generating layers), followed by layers which compute embedding distances, loss, and accuracy. **b** t-SNE visualization of the embeddings generated by the Triplet network from the X-ray images of the 29 bones. **c** Principle component analysis visualization of the embeddings generated by the Triplet network from the X-ray images of the 29 bones.

according to their real-world proportions improved the accuracy by 0.7%.

The following options had no effect on the Triplet accuracy: the specific layers used to achieve dimension reduction at the end of the base network, using the tertiary distance to compute loss (the distance between embeddings of positive and negative images), and reducing the number of classes from 29 to 24.

The Triplet network trained this way has the ability to generate embeddings of its input images which have the property that the dissimilarity between images translates into the Euclidean distance between their embeddings. The more dissimilar the images, the higher the Euclidean distance between their embeddings.

The triplet accuracy of 99.9% means that the network has learned very well to group together images in the Euclidean space that belong to the same classes, while keeping them apart from images belonging to different classes. This property made it possible to use the embeddings for classification purposes. The Triplet network architecture is shown in Fig. 3.

**Using bone fingerprints generated by the Triplet network for classification.** After extracting embeddings for each of the 26,100 bone images, a shallow classifier (a *k*-nearest neighbor classifier) was trained on these embeddings. The validation accuracy of this classifier was 100%, meaning that the classifier was able to assign these embeddings to their respective classes without any errors. This result was reproduced successfully with a support vector machine[44] classifier. It did not make a difference in accuracy whether the classifier was trained on embeddings from bones on which the neural network was also trained, or on completely new bones; we excluded five sample bones from the dataset, trained the Triplet network on the remaining 24 bones only, then generated embeddings for all the images of the five sample bones. Even after training the kNN classifier on the embeddings of the five excluded sample bones, the classifier still achieved a validation accuracy of 100%. This shows how well the Triplet network's ability to extract features characteristic to specific bones generalizes to previously unseen bones. Since the embeddings can be so accurately assigned to their classes (which represent specific

bones), each embedding can be thought of as a de facto unique fingerprint of a bone.

**Pairwise L2 fingerprint distances orient themselves on the margin value.** After establishing that bone image embeddings can be easily used to train a classifier that can tell different bones apart, we examined under which conditions randomly selected pairs of embeddings could be compared directly, i.e. without a classifier present, solely based on their distance in the Euclidean space. We conducted an experiment with the embeddings from images of the five bones the network had not been trained on; pairwise Euclidean (L2) distances were computed between all the images within the same class and between images of different classes. The results were then visualized in a scatter plot to show whether there was a distance threshold with all intra-class distances lying below it and all inter-class distances above it. This was not true of all the images. It does not, however, contradict the high kNN classifier accuracy, because classifying a group of images with a 100% accuracy only requires the classifier to find the best possible match for the input data and does not pose the additional constraint of how far in the Euclidean space the embeddings of the non-matching class instances must lie.

A clear threshold separating all intra-class embedding distances from all inter-class distances was found after embeddings of outlier images were removed. Such outliers were images of bones which depicted the bone at an angle that deviated strongly from either the anteroposterior or the mediolateral view. The observed separating threshold coincided with the margin value enforced during training, as was intuitively expected.

Figure 4 shows intra-class (orange) and inter-class (blue) embedding distances, separated by a gap in which the margin lies after the outliers are removed. When only embeddings from images are used where the depicted bone deviates by no more than 4° in either direction from the standard anteroposterior and mediolateral view and when the radiation energy used for the images is within the interval 146–158 keV, the margin value separates the inter-class and intra-class distances between embeddings with a 100.00% accuracy. Angle deviations have a stronger impact on this distance accuracy; when embeddings are

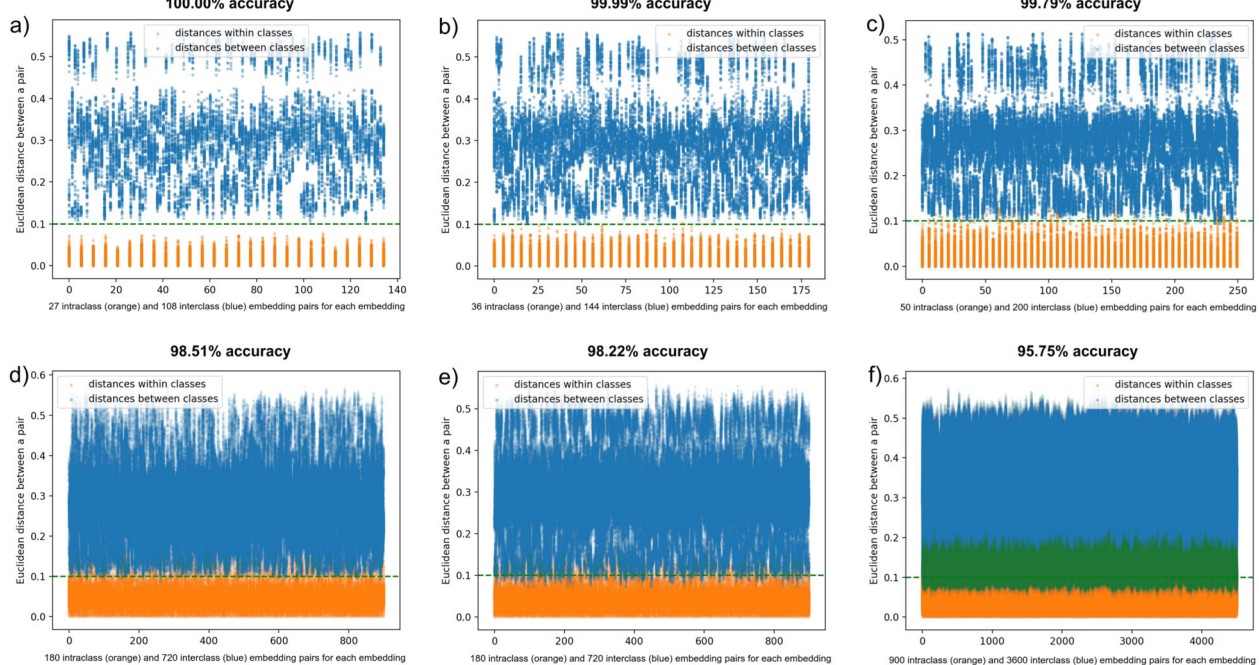

**Fig. 4 Distribution of pairwise Euclidean distances between embeddings of 4500 artificial X-ray images generated from five bones.** The Triplet network was not trained on any images of these bones. **a** Pairwise embedding distances between images depicting bones from an angle which did not deviate from the standard anteroposterior or mediolateral (AP/ML) view by more than 4° in any direction. Radiation energy interval: 146–158 keV. **b** Maximum deviation from AP/ML view: 4°. Radiation energy interval: 140–158 keV. **c** Maximum deviation from AP/ML view: 7°. Radiation energy interval: 140–158 keV. **d** Maximum deviation from AP/ML view: 22° around the bone's longitudinal axis and 4° around an axis perpendicular to image plane. Radiation energy interval: 140–158 keV. **e** Maximum deviation from AP/ML view: 4° around the bone's longitudinal axis and 22° around an axis perpendicular to image plane. Radiation energy interval: 140–158 keV. **f** Maximum deviation from AP/ML view: 22°. Radiation energy interval: 140–158 keV. Overlapping dots are marked green.

used which were generated from images encompassing the entire energy range used in generating the images (140–158 keV), the accuracy drops slightly to 99.99%. If both angles deviate by no more than 7°, we achieve an accuracy of 99.79%. If no more than one rotation angle deviates by no more than 22° (and the second one by no more than 4°), the accuracy is between 99.22 and 99.51%. If all generated images are used (angle deviations within a 22° interval, radiation energy interval 140–158 keV), the accuracy drops to 95.75%.

**Examination of the embedding space uncovers redundant dimensions**. In order to further examine the discrete embedding space, we took a balanced subset of bone images with 540 samples per class (these images comprised the training dataset of the Triplet network), grouped the $d$-dimensional embeddings generated from them by class, and calculated the per-class mean and standard deviation for each of the $d$ dimensions. The results are visualized in Fig. 5.

It became apparent that the Triplet network does not store the same amount of information in each dimension. There are dimensions which contain information for each class (e.g. dimension 8). A second group of dimensions store information only for certain classes (e.g. dimension 3). Thirty-five of the 128 dimensions are more than 99% sparse, i.e. they contain almost no information.

We tested the expressiveness of the embeddings with a very simple classification algorithm that classifies an embedding $X$ as belonging to class $C$ (defined by the $d$-dimensional mean vector $\mu_C$ and $d$-dimensional standard deviation vector $\sigma_C$) if $|\mu_C - X| < \sigma_C \cdot f$, where $f$ is a constant factor. That is, $X$ belongs to class $C$ if $X$ is (in most dimensions) near $\mu_C$ within a threshold of $\sigma_C \cdot f$.

**Table 1 Threshold factors ($f$) and corresponding classification accuracies using all embedding dimensions and a subset of less sparse embedding dimensions.**

| $f$ | Accuracy using all dimensions (%) | Accuracy without highly sparse dimensions (%) |
|---|---|---|
| 1.0 | 94.5 | 98.3 |
| 1.1 | 95.4 | 98.6 |
| 1.2 | 95.8 | 98.8 |
| 1.3 | 96.2 | 98.9 |
| 1.4 | 96.9 | 99.1 |
| 1.5 | 97.0 | 99.2 |
| 1.6 | 96.8 | 99.3 |
| 1.7 | 94.8 | 98.5 |
| 1.8 | 95.0 | 98.5 |
| 1.9 | 94.7 | 98.3 |
| 2.0 | 93.2 | 97.9 |

The test dataset consists of 4320 bone images (24 classes with 180 samples each) and their corresponding embedding vectors.

We first applied the classification algorithm to the complete test dataset and achieved 97% accuracy (Table 1, 2nd column). In a second experiment, we only evaluated dimensions that are less than 99% sparse, i.e. we dropped the 35 dimensions mentioned above. Surprisingly, we surpassed the accuracy achieved in the first experiment and achieved 99.3% classification accuracy (Table 1, 3rd column).

The examination of the embedding space shows that the embeddings generated by the Triplet network are highly

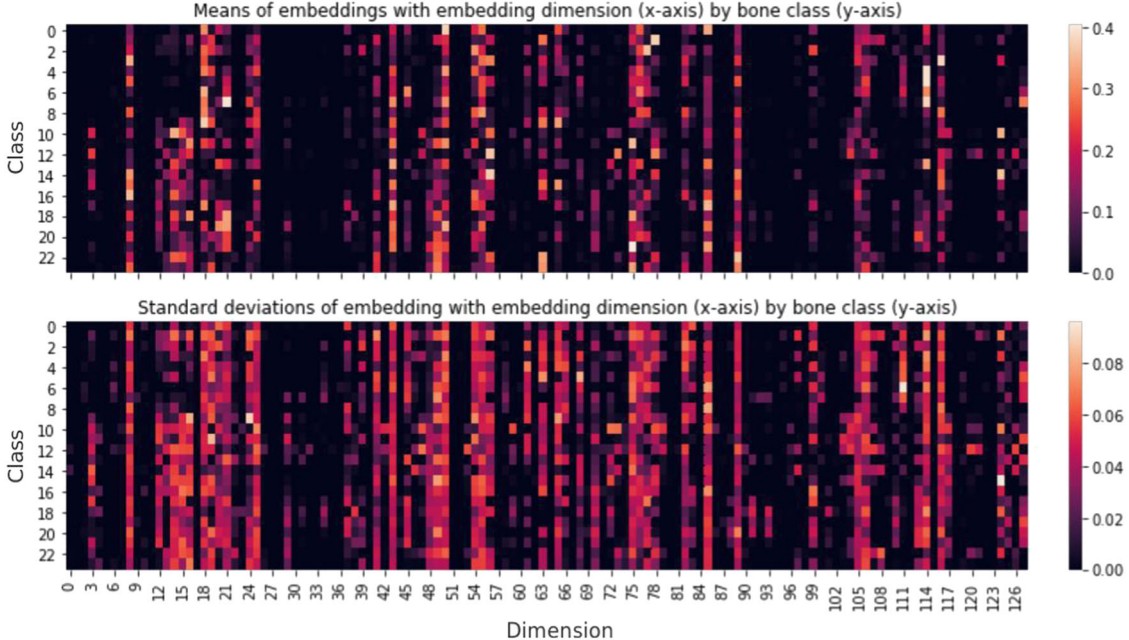

**Fig. 5 Statistical properties of the embedding space.** Visualization of means and standard deviations of 128-dimensional embeddings aggregated from 12,960 samples ($N = 1296$ samples belonging to 24 different cat femurs, with 540 samples per femur, each showing the bone under different conditions).

expressive; even a simple classification approach can achieve 99.3% accuracy. Interestingly, the classification results are better when the dimensions with low information density are dropped. This, of course, results in a loss of comparability between the reduced embeddings, and full, $d$-dimensional embeddings generated by the Triplet network for new classes.

**Bone shape estimation using a kNN classifier on Triplet network features.** We used the Triplet network-based on VGG-16 in combination with a kNN classifier to estimate the 3D shape of bones, given their 2D X-ray images. The 99.9% triplet accuracy and the 100% kNN accuracy mean that this neural network and kNN combination are excellent at predicting exact matches for bones, given their 2D images.

For shape estimation of bones unfamiliar to the network, however, high-quality nearest matches are required. We tested how well our Triplet network/kNN classifier combination performs when forced to make a prediction for the input image of a bone which was not part of the classifier's dataset, i.e. when any match predicted by our classifier will not be an exact match, but rather a nearest match, based on the features extracted by the network. How good these nearest matches are was tested by classifying image embeddings of the five sample bones excluded from the dataset the neural network was trained on, using the kNN classifier.

**Evaluation by comparing a bone match with other possible matches.** The evaluation of how well the predicted 3D shapes matched the true 3D shapes of the bones presented to the network as artificial 2D X-ray images was performed by computing the RMS distance and the Hausdorff distance between the predicted 3D shapes of each of the five sample bones and their true 3D shapes. The result was that on average, no more than 2.6 of the 24 bones would have been better matches. That is, only 10.8% of the bones the network had been trained on would have matched the shape of the bones in the input images better than the bones chosen by the classifier. In the best case, the best available match was predicted by the classifier. In the worst case, seven

**Table 2 Root mean square (RMS) distances and Hausdorff distances between the five sample bones S1–S5 and their matches predicted by the VGG-based Triplet network and kNN classifier on the 2D image dataset of bones scaled according to their real-world proportions.**

| Sample bone | Absolute RMS dist. (mm) | Absolute Hausdorff dist. (mm) | Relative RMS dist. | Relative Hausdorff dist. |
|---|---|---|---|---|
| S1 | 0.77 | 2.23 | 0.0065 | 0.0188 |
| S2 | 0.72 | 2.89 | 0.0066 | 0.0266 |
| S3 | 0.73 | 2.72 | 0.0058 | 0.0217 |
| S4 | 1.26 | 5.09 | 0.0114 | 0.0462 |
| S5 | 0.82 | 2.62 | 0.0061 | 0.0195 |
| S1–S5 average | 0.86 | 3.11 | 0.0073 | 0.0266 |

Absolute RMS distance is given in mm; relative distances are given with respect to the bounding box diagonal.

bones other than the one predicted would have been better matches. In the remaining three cases, one, two, and three bones would have been better matches.

**Evaluation by computing distances between predicted and true 3D shapes.** How good a 3D shape match was and also how many other candidates would have been better matches was evaluated by computing the RMS and Hausdorff distance between the predicted and the true 3D shape. For this purpose, the surface meshes of the bone from the input image and the predicted bone were aligned using the *MeshLab*[45] software and the RMS and Hausdorff distance between them was computed (also in *MeshLab*), both in millimeters and with respect to the bounding box diagonal. Table 2 summarizes the results.

Figure 6 shows the results of the qualitative evaluation—alignments between the 3D shapes predicted by the network–kNN combination and the true 3D shapes. For comparison, an alignment between two bones which are the worst possible pairwise match in the dataset is also shown.

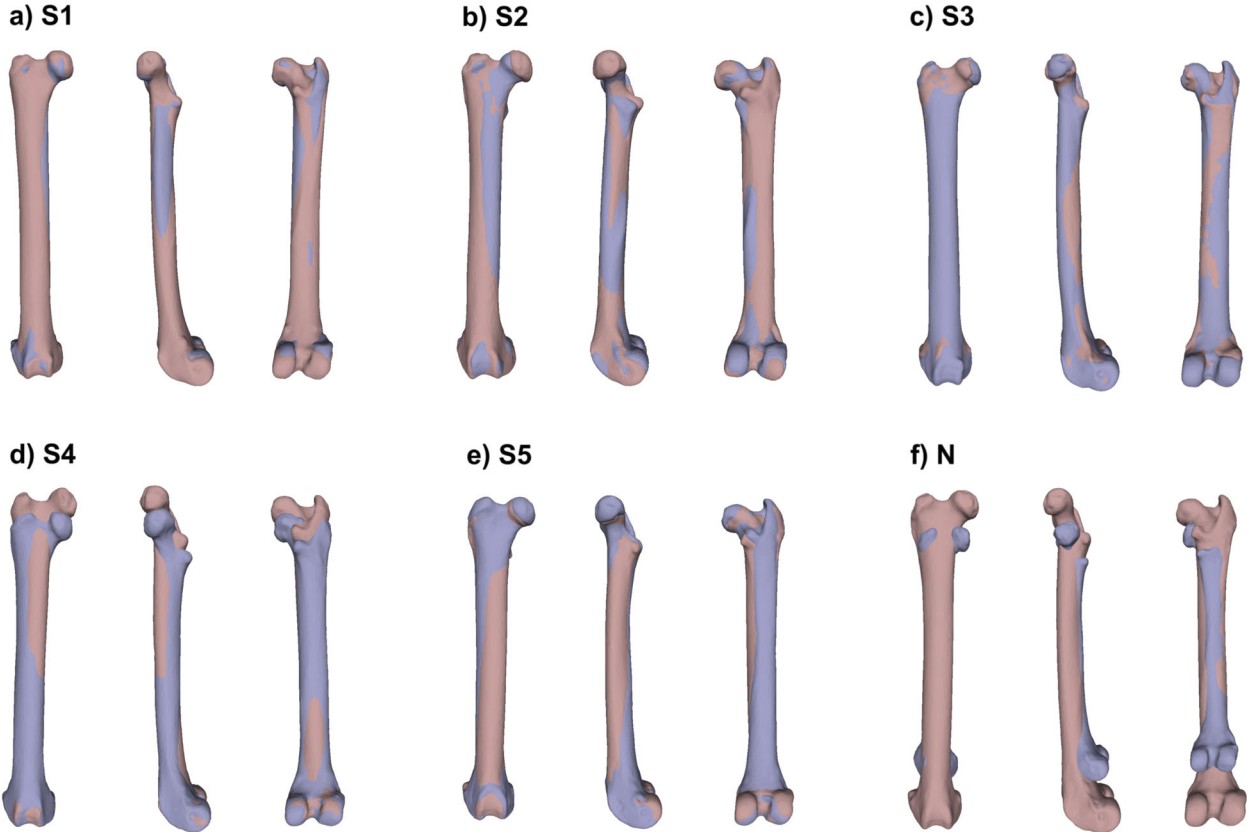

**Fig. 6 Qualitative assesment of shape prediction accuracy using alignments between true and predicted bone shapes. a–e** Alignments between the five sample bones S1–S5 and their closest matches as predicted by the VGG-based Triplet network in combination with the kNN classifier. **f** Alignment between the pair of bones which are the worst possible match present in the dataset, based on their RMS distance.

**Evaluation by comparing to SSMs and other approaches**. We also evaluated the quality of our 3D matches by comparing them with match quality achieved by SSMs implementation. For this purpose, we used three additional 3D CT scans that were not part of the original dataset of 29 bones. For each of the new bones, a 3D DICOM file as well as two corresponding natural X-ray images were available. The first natural X-ray image showed the bone from the anteroposterior and the second from the medio-lateral view. We let the network predict the closest 3D match for the natural X-ray image out of the original 24 bones. We then used the SSM software to generate a SSM from those 24 bones. The SSM model based on the 24 bones was then deformed by fitting it to the bones shown in the natural 2D X-ray image. The result of the fitting process was exported as a surface model and compared with the shape match predicted by our network/kNN classifier.

In two out of three cases, our network combined with the kNN lead to a better match. In the third case, the SSM produced a better match. The RMS distances between the predicted and true shapes when using our method were 0.89, 2.53, and 0.93 mm. The RMS distances when using the SSM method were 1.35, 3.07, and 0.80 mm, respectively. It is notable that in the case of the second bone, where both the SSM and our model achieved an unusually high RMS error, and also in the case of the third bone, where our model performed worse than the SSM approach, the bone in the natural X-ray image was approx. 6% shorter in the anteroposterior view than in the lateral view, an issue that occurs in practice due to difficulties with positioning the patient during X-ray image acquisition in the frontal plane. Our network was, however, not trained on bones with different lengths in the two views, resulting from positioning errors around the $z$-axis in the anteroposterior

views. A source of error which could easily been taken into account when re-training the network with artificial 2D X-ray images simulating positioning error along all three axes.

A comparison with other approaches to 2D to 3D bone shape estimation found through literature research shows that our approach can compete with them; for the eight bones we examined (five sample bones and three bones paired with their natural 2D X-ray images), the average RMS distance between them and their match predicted by the Triplet network/kNN combination was 1.08 mm (0.89 mm for the five sample bones where predictions were made based on artificial X-ray images and 1.45 mm for the three bones where predictions were made from natural X-ray images).

In comparison, the mean or RMS distance between predicted and true shapes achieved by eight other examined approaches was, on average, 1.32 mm. The examined approaches are a Laplacian surface deformation method (1.2 mm error), a fast Fourier transform-based method (1.4 mm error), a method based on iterative nonrigid 2D point-matching process and thin-plate spline-based deformation (0.9 mm error), a method using a hybrid atlas bone fitted to three 2D projection images (2 mm error), a non-stereo corresponding contour method (1.4 mm error), and three different SSM-based methods (1, 1.68, and 1 mm error)[1].

**Evaluation on deformed bones**. An examination of the properties of embeddings generated for deformed bones yielded the following results.

As we see in Fig. 4, the Euclidean distance between image embeddings of different classes using a margin value of 0.1 ranges

between approx. 0.06 and 0.6 if all images are considered and between 0.11 and 0.6 if only images are considered that do not deviate in their angle and illumination too strongly from the average. These are all image embeddings of the five sample bones S1–S5, without any deformities.

For comparison, we introduced deformities to the images of the sample bones S1–S5 either by manually removing a part of the bone shaft to make it appear unnaturally thin (Fig. 1e) or by blending the depicted bone together with another image of the same bone, the second image having been rotated in such a way that the resulting merged bone appeared to have its proximal end broken off at an angle (Fig. 1f).

We then generated embeddings for images of these deformed bones and observed the Euclidean distance between these embeddings and those of the images of the healthy sample bones S1–S5 from which each deformed bone was generated.

During our experiments with deformities that covered up a part of the bone, the Triplet network was on average able to correctly assign the image of the deformed bone to 92.59% of the images of the non-deformed, healthy version of the same bone. The deformed bone image was also compared to all of the images of the other four healthy bones, and not a single match was found, i.e. the false-positive rate was 0%.

The number of correct matches varied from sample bone to sample bone; for the five bones S1, S2, S3, S4, and S5, the percentage of their images to which an unnaturally thin, deformed bone could be correctly assigned was 96.30%, 100%, 92.60%, 100%, and 74.07%, respectively. The one bone image with the proximal end broken off at an angle could only be matched to 57.14% of the images of the healthy, original bone. However, even in this case, the only errors were false negatives, i.e. the prediction that two bones belong to the same class was always reliable, when it was made. Only when the network failed to find a match was it still possible that the match existed.

We therefore conclude that the Triplet network's prediction of two bones belonging to the same class can be considered reliable, even if deformities are present. False identification of a bone as a match if the bone was not a true match has not occurred for any of the deformed images. However, the more prominent the deformity, the more likely the network is to overlook an exact match (a match between a deformed bone and its healthy original), even if a match is present.

Notably, even though the Triplet network never wrongly matches examples of different classes, it is not possible to differentiate between the deformed bones and healthy bones that belong to different classes, based on their embeddings alone. Their pairwise Euclidean distances stay within the [0.11, 0.6] range, as is the case with healthy bones belonging to different classes. This can be explained by the fact that the triplet loss function was not designed to enforce any restrictions on the positioning of non-matching examples in the embedding space, such as increasing the Euclidean distance linearly with bone dissimilarity.

The 3D shape prediction deteriorates visibly (by approx. 25.83%) in the presence of deformities. A comparison between the closest matches predicted for the deformed bones and the closest matches predicted for their healthy counterparts (the original bones from which the deformed bones were generated) shows that while for the healthy original bones, 2.6 out of the 24 bone candidates would have been better shape matches on average, for the deformed bones, 9.0 out of the 24 possibilities would have been better shape matches (a deterioration by 25.83%).

It should also be noted when considering bone deformities that two out of the three additional bones used for validation that are mentioned above, i.e. those with both a 3D CT scan and a natural 2D X-ray image, can be considered deformed from the viewpoint of our network, since their natural X-ray images differed in their lengths by approx. 6% in the anteroposterior and mediolateral views. This resulted in a higher RMS distance in the first case (2.53 mm vs. our average of 1.08 mm) and in a RMS distance which was 0.13 mm higher than the RMS distance achieved using the SSM approach in the second case. This illustrates that suboptimal input image quality affects the quality of the shape predictions to a high degree.

## Discussion

In this paper, we presented multiple ways in which deep neural networks are capable of working with X-ray images of bones. Both transfer learning and triplet loss training in combination with a kNN classifier showed that neural networks pre-trained on the ImageNet dataset and fine-tuned on a dataset of artificially generated bone X-ray images are able to differentiate between femurs from different cat specimens with an accuracy of 100.0%. This result demonstrates neural networks' ability to extract meaningful features from X-ray images and opens up new possibilities for deep-learning-based, fully automated work with X-ray images of bones.

We tested our triplet loss-trained neural network (combined with a kNN classifier) on images of eight bones to determine their 3D shape by selecting the best-fitting matches out of a set of 24 choices. Our network first extracted a 128-dimensional embedding from each input image, and a kNN classifier then determined the best match based on embedding distances in the Euclidean space. The average root mean squared distance between our shape predictions and the ground truth was 1.08 mm.

The comparison with existing 2D to 3D bone shape estimation approaches shows that our approach can compete with them and even seems to perform better than the average achieved by eight other approaches we examined[1], even though an exact comparison is not possible, since the respective publications use absolute shape distances in mm, not relative ones (normalized by their bounding box diagonal).

A clear advantage in comparison with other methods such as SSMs is that our approach is completely domain-agnostic and as such does not require any previous knowledge about the bones in our dataset, such as their geometry. Re-training our classifier to use completely different bones is as easy as supplying it with a new set of 3D CT scans. After a training dataset of images has been generated, our network learns to extract all features necessary for bone classification and shape estimation automatically.

A very important finding is that a neural network trained using the triplet loss method is able to determine the identity of a bone based only on its 2D X-ray image. The 128-dimensional embedding our network generates for each input image can either be used for shape estimation by letting a kNN classifier determine the nearest match for an unknown bone, or for determining the identity of a bone. Determining bone identity is possible either by using a kNN classifier trained on embeddings of bones that are possible matches for the bone in question, or by pairwise comparisons directly, using the triplet margin as the minimum Euclidean distance between embeddings of the bone images being compared. When using pairwise comparisons, we achieved a 96% accuracy for a total of 4500 X-ray images generated from 3D CT scans of five bones the Triplet network had not been trained on. The 96% accuracy was achieved when the rotation angle varied within a 22° interval. A 100% accuracy was achieved when the rotation angle was restricted to an 8° interval. In the presence of deformities, the Triplet network is even capable of finding a healthy bone match if the healthy bone was entered into the database before the deformity occurred.

However, the quality of the shape estimation deteriorates if prominent deformities are present.

Even though the shape prediction accuracy deteriorates when deformities are introduced into the bone images, the bone identity prediction based on image embeddings retains a 0% false-positive rate, i.e. even a deformed bone can still be correctly assigned to the original bone before deformation, as long as their viewing angles lie in approx. the same interval (7°). The more strongly the bone is deformed, the less likely it is the network can match it with any bone at all, even if the original bone is present.

In addition, our findings about bone fingerprinting have interesting implications for database systems—such fingerprints may be used to either enrich bone databases by highly compressed information about the bones, or they can even be used as a search key, even if the depicted bone was deformed slightly since having been entered into the database. This has possible forensic applications where image-based look-up of bones with unknown origin in bone databases would be sped up and could be performed within seconds.

## Methods

**Dataset**. 2D and 3D data: Our dataset consisted of 29 3D CT scans of femurs of 29 different cats in the DICOM format. Using the software *MeVisLab*, we generated 900 artificial 2D X-ray images for each of the 29 CT scans. For this purpose, we created a *MeVisLab* macromodule which contained a Python script. In this script, we set the values for rotation around the *X*-axis to be within the interval [70, 112] and sampled every third value from this interval: 70, 73, 76,…, 112. For the rotation around the *Y*-axis, the interval was set to [−21, 22] and we also sampled every third value. For the radiation energy, we sampled every sixth value from the interval [140, 161]. We sampled these values in a triple nested loop for each DICOM input file, which resulted in $15 \times 15 \times 4 = 900$ different value combinations. The rotations around the two axes rotated the bone slightly around its own longitudinal axis as well as around an axis perpendicular to the 2D image plane. In the innermost of the nested loops, the sampled values were assigned to the *DRR* and *DRRLUT* module fields of our *MeVisLab* network, after which the resulting image was saved in the PNG format using the *ImageSave* Module. The resulting images were slightly blurred using a sigma value of 1.0 to remove minor CT artifacts[46]. This way, we automatically generated 26,100 images from the 29 DICOM files.

These 26,100 images served as the dataset to train and evaluate our neural network/kNN classifier on.

Standard viewing angles: For each DICOM file, we used *OrthoSwapFlip MeVisLab* module to rotate the bone by 90°. After this, we had 29 DICOMs showing a bone from the anteroposterior and 29 from the mediolateral view. This way, we could more easily generate 900 $400 \times 800$ pixel images with the anteroposterior and 900 $400 \times 800$ pixel images with the mediolateral view. We then merged these two groups, creating a dataset of 900 $800 \times 800$ pixel images per class, each showing the same bone from the two orthogonal angles next to each other. This resolution is higher than many CNNs accept; this was on purpose so that no upscaling would be needed for the networks.

Bone image scaling: We also created a version of our 2D image dataset which contained images of bones scaled according to their real-world proportions. This dataset version achieved slightly higher Triplet network accuracy, and a much higher accuracy when 3D shapes were predicted. The bones depicted in the images were scaled, without modifying the image pixel resolution, the following way: A scaling object with a known real-world size (10 mm) was drawn into all DICOM volumes using the *MeVisLab* module *DrawVoxels3D*. 2D images were then generated from these modified DICOM files, one image from the anteroposterior DICOM file of a bone and the second image from the corresponding mediolateral DICOM file of the same bone. The reason was that the DICOM volumes are not cubic, nor was the bone always positioned perfectly in the middle, which is why the anteroposterior and mediolateral DICOM volumes had to be scaled separately. The pixel sizes of the scaling objects next to all bones were measured and scaling coefficients were calculated to determine by how many percent the content of each image had to be shrunk to make the scaling objects in all images have the same pixel size. The images of the entire 2D image dataset were then shrunk according to these coefficients. Anteroposterior and mediolateral views of the same image could have different coefficients and had to be shrunk separately. After shrinking, each image was padded with black borders (the background color) to bring their resolution back to $400 \times 800$ pixels.

Surface model extraction for evaluation: For evaluation purposes, the surface model (mesh) of each 3D CT scan was extracted so that the difference between a predicted shape and the true shape could be measured. First, a mesh in the STL format was extracted from each DICOM file using the *MeVisLab* modules *WEMIsoSurface* and *WEMSave*. Setting the Iso value in *WEMIsoSurface* to 1370 led to successful removal of the soft tissue of the DICOM files, so that only the bone tissue remained and could be exported as a mesh. This mesh was then post-processed

to remove the inner bone structure as well as the neighboring bones (cat pelvis, tibia, and patella). The post-processing was done in the *Materialise Mimics* software. A viable alternative is using the free software *MeshLab* and the freely available *Microsoft 3D Builder*. In *MeshLab*, the bone artifacts can be selected both manually (for removing the neighboring bones) and by the degree of their occlusion (for removing the inner bone parts). Closing the holes caused by removal of neighboring bones can be done when importing the meshes into *Microsoft 3D Builder*—the software merely asks whether any mesh holes should be closed before importing. However, *Materialise Mimics* lead to better results than the free alternatives.

Natural X-ray image pre-processing: The natural X-ray images used for comparison to SSM were pre-processed approximately the same way the artificial X-ray images were cropped—closely enough that only a minimum of the neighboring bones was visible. These X-ray images were made with a scaling object of known real-world size present on the X-ray table, so that they could be scaled in proportion to the artificial X-ray images on which the network had been trained.

**Statistics and reproducibility**. The split of the 2D images into a training (70%) and a validation (30%) set for the Triplet network was random and remained without change for all experiments. Each of the experiments examining the influence of different hyperparameters (or dataset sizes, or other factors) on the Triplet accuracy was repeated at least three times to ensure the accuracy stayed within a 0.2% range. The initial selection of the image triplets for the first training epoch of each experiment was deterministic and identical for each experiment; the triplet selection for subsequent epochs depended on what the Triplet network learned in the first epoch and differed for each training. The order in which the experiments were conducted was random. Data pre-processing steps and sample sizes used are described in "Methods", under "Dataset".

**Neural network architecture**. The triplet loss training was implemented by building all the necessary computations directly into the neural network architecture. We used first the neural network ResNet-50 (ref. [37]) and then the network VGG-16 (ref. [38]) as the base building blocks of the Triplet network.

ResNet-50 (ref. [37]) is a 50-layer deep CNN first published by in 2015 by He et al. It achieves a 6.71% top-5 error rate on the benchmark ImageNet[35] dataset. Each layer in the network extracts a new layer of features, with a classifier at the end. The number of layers has been shown to be of crucial importance. However, stacking too many layers, even if the problem of exploding/vanishing gradients is taken care of, leads to a performance degradation. He et al. solve this problem by making heavy use of so-called shortcut (residual) connections between the layers of their ResNet networks. This way, networks of over 150 layers have been successfully trained. We used the 50-layer deep variant of the ResNet architecture as basis for our feature extractor within the Triplet network, on the assumption that the features present in the X-ray images might be complex enough and thus require such a large number of abstraction layers.

VGG-16 (ref. [38]) is a 16-layer deep convolutional network published by Simonyan and Zisserman in 2014. It achieves a 9.33% top-5-error rate on the benchmark ImageNet dataset. What distinguishes it from its predecessors is that it successfully trades larger convolutional filters for smaller ones stacked behind one another along the length of the network, effectively covering a larger receptive field using a smaller number of network parameters. Similarly to ResNet, it also performed a large increase of the number of its layers when compared to its predecessors. However, with 16 layers, it is much shorter than ResNet-50, and its architecture is also different. We used VGG-16 in later experiments to cover the case that the features in the X-ray images might be too simple for ResNet-50 and a more shallow network might extract features better suited for shape estimation. Our experiments showed that this was, in fact, the case. While the triplet accuracy was only slightly improved by using VGG-16 instead of ResNet-50 (by 0.1%), the 3D shape predictions were improved by 4.2%. Using ResNet-50, an average of 3.6 out of 24 bones would have been better shape matches for out five sample bones. When using VGG-16, 2.6 of the bones not predicted by our classifier would have been better matches.

The Triplet network used to generate embeddings of the input X-ray images was built the following way: It consisted of three identical copies of the so-called base network. ResNet-50, in later experiments VGG-16, in combination with dimension-reducing layers, was used as the base network. Each of the three identical base network copies received a different triplet image as input data: One base network received the so-called anchor image (which was randomly selected from the training set), the second base network received the positive image (image from the same class as anchor), and the third one received the negative image (image from a different class than anchor). The base networks themselves had to be modified, because they had to be used in a capacity as feature extractors, not classifiers. To achieve this, we removed their last (softmax classifier) layer and kept the networks only up to their last fully connected layer, which produced an embedding with over 1000 dimensions. We then reduced this output down to $d$ dimensions, with $d$ set to 128, 64, 32, and 16 during our experiments. We performed this dimension reduction by appending three pairs of layers—a dimension-reducing fully connected layer followed by an L2 regularization layer. The last of the three fully connected layers produced the $d$-dimensional output, and the last L2 layer normalized it to have unit length. The layer that followed combined the embeddings by stacking them together so that they would be available for the accuracy and loss functions.

The triplet loss was defined to be zero when the difference between the Euclidean distance of anchor and positive embedding and the Euclidean distance of anchor and negative embedding was larger than margin, as shown in Eq. (2).

**Reporting summary**. Further information on research design is available in the Nature Research Reporting Summary linked to this article.

## Data availability

The datasets generated during and/or analyzed during the current study (DICOM, PNG/JPG, and STL files) are available in the Zenodo repository[47], including the training dataset containing 2D images of the 24 bones used by the Triplet network and kNN classifier. This training dataset of 2D images is also available as Supplementary Data 1.

## Code availability

We declare that the programming code necessary to reproduce the findings of this paper is available at Zenodo[48] (code for data pre-processing, training the Triplet neural network and kNN classifier, and for performing inference, as well as the fully trained Triplet network and kNN classifier itself), and within the Supplementary software files. Other deep-learning models reported in this paper are publicly available in TensorFlow and on GitHub.

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

## Acknowledgements

We gratefully acknowledge the support of the Freie Universität Berlin where this research was conducted, as well as the support by the Open Access Publication Fund of the Freie Universität Berlin. We gratefully acknowledge the support of Shapemeans GmbH which provided the software used to generate the statistical shape model from the 24 bones and software for fitting this model to 2D X-ray images.

## Author contributions

J.C. conceived the research, implemented automated data processing and neural network training, conducted most experiments, and wrote the manuscript. J.P. conducted the pairwise embedding distances experiments and wrote the corresponding manuscript section. D.M. designed and conducted further examination of the embedding space and wrote the corresponding manuscript section. P.B. provided and pre-processed the data and gave input for the manuscript. N.J.L., A.V., and P.B. supervised the research.

## Competing interests

The authors declare no competing interests.
