## [Peer Review File · Communications Biology]

Reviewers' comments:

Reviewer #1 (Remarks to the Author):

The paper presents a new interesting method to estimate 3D bone structures for a pair of 2D X-ray images. The method claims not to rely on any previous knowledge about bone geometric shapes. Triplet loss-trained deep neural network is used to identify the most suitable 3D bone shapes from a pre-defined set of data. The evaluation of the proposed method has been conducted and the average root mean square distance of 1.08 mm between the reconstructed and the ground truth shapes is reported in comparison to the accuracies published in the other works.

In general, I am happy with the method and results presented in the paper. However, there are some aspects to be addressed. Hence, I would recommend the paper for publication subject to the following points being addressed:

1. Since 2017, there have been a large number of publications on using deep neural networks or deep learning to reconstruct or infer 3D shapes from 2D images. Many of those methods could potentially, probably with some modification, be used with X-ray images. The authors should survey and review more of those works in the paper. Currently, most publications mentioned are related to X-ray only.
2. Triplet loss training method is used to train the CNN. As currently, there is a huge variety of training methods available, it would be a good idea for the authors to present the justification to use this method in more detail, especially, in comparison with other training methods.
3. Are there any specific criteria used to choose the 29 cats femurs to ensure that the 29 femurs appropriately represent the all relevant shape variation? Since all the training data is generated from those 29 femurs, it is quite important to consider this aspect.
4. In reality, there may be some deformed bones which are quite different from the shapes in the training data. How would the method cope with that?
5. As mentioned above, in the evaluation section, the authors may want to mention other 3D reconstruction using deep learning methods (see 1. above) in comparison to the proposed method.

Reviewer #3 (Remarks to the Author):

This paper introduces an approach to compare 2D X-ray images and 3D models for the task of bone classification. The specific algorithm takes 2D X-ray images of a bone as input, and finds the closest bone in a 3D database. The retrieved bone serves as the reconstruction. This retrieval procedure is treated as a classification problem, where one database shape is associated with one class. The classification leverages a neural network that is trained using a triplet loss.

Although the idea of using cutting-edge machine learning techniques for medical imaging is interesting. This paper does not present any technical contributions. It is more an application of existing techniques. There are a couple of things to improve on:

- 1) The reconstruction uses a database shape. It would be interesting to see if applying a non-rigid deformation can further improve the results.
- 2) It would also be interesting to learn a shape space of bones and use a neural network to determine the best matching latent parameter of the shape in this shape space.

In summary, although the paper presents an interesting application. The proposed approach lacks technical contributions. The reconstruction results are not sufficiently evaluated.

Dear editor and dear reviewers,

thank you for taking the time to review our work!

We tried to do our best to address the concerns you raised in the form of our revised manuscript. All changes we made are highlighted in the manuscript itself. This document lists the new sections point by point, in a question-answer manner. We hope we could answer everything to your satisfaction and are looking forward to your feedback.

Best,

Jana Cavojska

Responses to reviewer #1's remarks:

- 1. Since 2017, there have been a large number of publications on using deep neural networks or deep learning to reconstruct or infer 3D shapes from 2D images. Many of those methods could potentially, probably with some modification, be used with X-ray images. The authors should survey and review more of those works in the paper. Currently, most publications mentioned are related to X-ray only.**
 - 5. As mentioned above, in the evaluation section, the authors may want to mention other 3D reconstruction using deep learning methods (see 1. above) in comparison to the proposed method.**
-

Our summary of other deep learning-based 3D shape reconstruction approaches can now be found in the Introduction:

“In recent years, many deep learning-based approaches to 3D object reconstruction from 2D images emerged [11] [12] [13] [14] [15] [16] [17] [18] [19] [20].

The approach by Henzler et al. [12] uses a neural network to generate a 3D shape from a single bone radiograph, with the goal to recover 3D data for databases of fossils where only 2D data is available. The main drawback of this method which operates in the absence of any priors is that it can generate implausible 3D shapes, such as skulls that do not optically resemble skulls.

Other methods exist that attempt to reconstruct the 3D shape of an object from a single image [11], using for example segmentation masks and key-points to build category-specific shape models [13] or using surface normal prediction [14] or mesh reconstruction while exploiting shading and lighting information [15]. However, in the absence of either multiple viewpoints or previous knowledge about object geometry, no accurate reconstruction of the occluded parts of the object can be guaranteed.

The recurrent 3D-R2N2 network by Choy et al. [16] learns a mapping from observations to the underlying 3D shapes of objects from a large collection of training data. The network first generates an embedding of the 2D image and then reconstructs the object in the form of a 3D occupancy grid based on this embedding. A single input image is sufficient for the 3D reconstruction; however, if multiple images of the same object from different views are available, they are used to refine the initially estimated 3D shape. This method suffers from the problem

that when a set of 2D input images is fed into the network in a different order, it produces different reconstruction results [17].

Xie et al. [17] address this and other problems using their Pix2Vox framework for single-view and multi-view 3D reconstruction. Their encoder-decoder first generates a coarse 3D volume from each input image. Then, a context-aware fusion module adaptively selects high-quality reconstructions for each part (e.g., table legs) from different coarse 3D volumes to obtain a fused 3D volume. Finally, a refiner refines the fused 3D volume to generate the final 3D output. Both 3D-R2N2 and Pix2Vox learn a discrete embedding space from which the 3D shapes are reconstructed. Others [18] [19] propose continuous embedding spaces by incorporating the capabilities of a variational autoencoder into their pipeline.

Wu et al. [18] propose a 3D-VAE-GAN architecture where a GAN (generative adversarial network) is trained to generate the 3D shape from a latent space, and a VAE (variational autoencoder) is used to ensure the latent space is continuous.

Liu et al. [19] work with a hierarchical continuous latent space, meaning that instead of using a single embedding vector as the intermediate representation, they generate a more complex internal variable structure consisting of one global latent variable layer hardwired to a set of local latent variable layers, each representing one level of feature abstraction. This more complex structure aims to improve the quality of the GAN reconstruction and to prevent the blurriness of the reconstructed images.

We present...

The existing deep-learning based methods that generate an embedding of the 2D image as an intermediate representation [16] [17] [18] [19] optimize their embeddings to hold 3D shape information, while our bone embeddings are optimized to uniquely identify highly similar 3D objects, while still making 3D shape inference possible.

Jointly training one of the networks that relies on embeddings with our network can therefore result in embeddings which combine these capabilities and would be a very interesting future application. It remains to be seen whether the pairs of orthogonal images in our bone dataset contain sufficient information for such a shape reconstruction and whether the generated 3D shapes are suitable for clinical purposes.”

2. Triplet loss training method is used to train the CNN. As currently, there is a huge variety of training methods available, it would be a good idea for the authors to present the justification to use this method in more detail, especially, in comparison with other training methods.

We have expanded the section “Classification vs. feature extraction using a neural network” that

explains our motivation for using triplet loss. The newly added text reads as follows:

“The reason why we chose the triplet loss method over transfer learning was that we were looking for a way to extract features from the input data that could serve purposes other than simple classification. We decided to use triplet loss over similar distance metric learning approaches such as contrastive [29] or magnet [30] loss because recent publications [12] [13] have shown how successful it is in extracting visual features from large datasets, even for tasks such as face recognition, as demonstrated by the 99.63% accuracy the FaceNet network achieves on the widely used Labeled Faces in the Wild (LFW) dataset and the 95.12% accuracy on the YouTube Faces Database [12]. Some publications argue for using softmax combined with metric learning [13], or for using magnet loss [30] over triplet loss, while others defend triplet loss against these approaches [13] [31]. A drawback of the representations learned through softmax is that they attain limited intra-class compactness and inter-class separation when compared to triplet embeddings [31]. In addition, the publication defending magnet over triplet loss verifies their claims on datasets of no more than 120 classes, whereas Schroff et al. [12] use their FaceNet triplet network on 1595 classes in case of the YouTube Faces Database, and 5749 classes in case of the LFW database, achieving the abovementioned high accuracies. This was a strong argument for us to use triplet loss, because it indicates the robustness of this loss function at scale (for thousands of classes) if our approach was to be expanded to use a much larger amount of different bones in the future. One possible use case can then be to augment large forensic bone databases with embedding data, so that each bone could be searched for using only its image embedding.”

3. Are there any specific criteria used to choose the 29 cats femurs to ensure that the 29 femures appropriately represent the all relevant shape variation? Since all the training data is generated from those 29 femures, it is quite important to consider this aspect.

“We chose to work with a dataset of 29 bones because it provides a good compromise between roughly representing the variety of feline femurs that are commonly encountered at clinics (consisting of specimens which differ in their lengths, widths and other shape variations), and being small enough that it is easy to generate, should the need arise to expand our approach to other bones than femurs, with a comparable accuracy.”

4. In reality, there may be some deformed bones which are quite different from the shapes in the training data. How would the method cope with that?

See subheading “Evaluation on deformed bones”, as well as Figure 1d):

“An examination of the properties of embeddings generated for deformed bones yielded the following results:

As we see in Figure 4, the Euclidean distance between image embeddings of different classes using a margin value of 0.1 ranges between approx. 0.06 and 0.6 if all images are considered and between 0.11 and 0.6 if only images are considered that do not deviate in their angle and illumination too strongly from the average. These are all image embeddings of the five sample bones S1-S5, without any deformities.

For comparison, we introduced deformities to the images of the sample bones S1-S5 by either manually removing a part of the bone shaft to make it appear unnaturally thin (Figure 1 d) middle), or by blending the depicted bone together with another image of the same bone, the second image having been rotated in such a way that the resulting merged bone appeared to have its proximal end broken off at an angle (Figure 1 d) right).

We then generated embeddings for images of these deformed bones and observed the Euclidean distance between these embeddings and those of the images of the healthy sample bones S1-S5 from which each deformed bone was generated.

During our experiments with deformities that covered up a part of the bone, the Triplet network was on average able to correctly assign the image of the deformed bone to 92.59% of the images of the non-deformed, healthy version of the same bone. The deformed bone image was also compared to all of the images of the other four healthy bones, and not a single match was found, i. e. the false positive rate was 0%.

The number of correct matches varied from sample bone to sample bone; for the five bones S1, S2, S3, S4, S5, the percentage of their images to which an unnaturally thin, deformed bone could be correctly assigned was 96.30%, 100%, 92.60%, 100%, 74.07%, respectively. The one bone image with the proximal end broken off at an angle could only be matched to 57.14% of the images of the healthy, original bone. However, even in this case, the only errors were false negatives, i.e. the prediction that two bones belong to the same class was always reliable, when it was made. Only when the network failed to find a match was it still possible that the match existed.

We therefore conclude that the Triplet network's prediction of two bones belonging to the same class can be considered reliable, even if deformities are present. False identification of a bone as a match if the bone was not a true match has not occurred for any of the deformed images. However, the more prominent the deformity, the more likely the network is to overlook an exact match (a match between a deformed bone and its healthy original), even if a match is present.

Notably, even though the Triplet network never wrongly matches examples of different classes, it is not possible to differentiate between the deformed bones and healthy bones that belong to different classes, based on their embeddings alone. Their pairwise Euclidean distances stay within the [0.11, 0.6] range, as is the case with healthy bones belonging to different classes. This can be explained by the fact that the triplet loss function was not designed to enforce any restrictions on the positioning of non-matching examples in the embedding space, such as increasing the Euclidean distance linearly with bone dissimilarity.

The 3D shape prediction deteriorates visibly (by approx. 25.83%) in the presence of deformities. A comparison between the closest matches predicted for the deformed bones and the closest matches predicted for their healthy counterparts (the original bones from which the deformed bones were generated) shows that while for the healthy original bones, 2.6 out of the 24 bone candidates would have been better shape matches on average, for the deformed bones, 9.0 out of the 24 possibilities would have been better shape matches (a deterioration by 25.83%).

It should also be noted when considering bone deformities that two out of the three additional bones used for validation that are mentioned above, i. e. those with both a 3D CT scan and a natural 2D X-ray image, can be considered deformed from the viewpoint of our network, since their natural X-ray images differed in their lengths by approx. 6% in the AP and ML views. This resulted in a higher RMS distance in the first case (2.53mm vs. our average of 1.08mm) and in a RMS distance which was 0.13mm higher than the RMS distance achieved using the SSM approach in the second case. This illustrates that suboptimal input image quality affects the quality of the shape predictions, to a significant degree.”

Responses to reviewer #3's remarks:

1) The reconstruction uses a database shape. It would be interesting to see if applying a non-rigid deformation can further improve the results.

Unfortunately, this would have been problematic, as we do not see a way to modify the image-based neural network approach the results of which we presented in order to meet this requirement. Our image-based classification is a completely different approach than non-rigid deformation of a bone model would be, and it was not our aim to present the results of non-rigid deformation (for which many publications already exist), but rather to present the outcome for a completely domain-agnostic image-based approach, which additionally makes it possible to uniquely identify the input data by extracting low-dimensional embeddings from it.

Applying non-rigid deformation would pose many additional constraints on the presented approach, such as previous knowledge about bone geometry at the very least, which would make the findings of our paper more cumbersome to use in practice, rendering them less helpful and less relevant.

2) It would also be interesting to learn a shape space of bones and use a neural network to determine the best matching latent parameter of the shape in this shape space.

See subheading “Examination of the embedding space”, as well as Figure 5 and Table 1:

“In order to further examine the discrete embedding space, we took a balanced subset of bone images with 540 samples per class (these images comprised the training dataset of the Triplet network), grouped the d-dimensional embeddings generated from them by class and calculated the per-class mean and standard deviation for each of the d dimensions. The results are visualized in Figure 5.

It became apparent that the Triplet network does not store the same amount of information in each dimension. There are dimensions which contain information for each class (e.g. dimension 8). A second group of dimensions store information only for certain classes (e.g. dimension 3). 35 of the 128 dimensions are more than 99% sparse, i.e. they contain almost no information. We tested the expressiveness of the embeddings with a very simple classification algorithm which classifies an embedding X as belonging to class C (defined by the d-dimensional mean vector μ_c and d-dimensional standard deviation vector σ_c) if $|\mu_c - X| < \sigma_c \cdot f$, where f is a constant factor. I.e. X belongs to class C if X is (in most dimensions) near μ_c within a threshold of $\sigma_c \cdot f$.

The test dataset consists of 4320 bone images (24 classes with 180 samples each) and their corresponding embedding vectors.

We first applied the classification algorithm to the complete test dataset and achieved 97% accuracy (Table 1, 2nd column.). In a second experiment, we only evaluated dimensions which are less than 99% sparse, i.e. we dropped the 35 dimensions mentioned above. Surprisingly, we surpassed the accuracy achieved in the first experiment and achieved 99.3% classification accuracy (Table 1, 3rd column.).

The examination of the embedding space shows that the embeddings generated by the Triplet network are highly expressive; even a simple classification approach can achieve 99.3% accuracy. Interestingly, the classification results are better when the dimensions with low information density are dropped. This, of course, results in a loss of comparability between the reduced embeddings, and full, d-dimensional embeddings generated by the Triplet network for new classes.”

In summary, although the paper presents an interesting application. The proposed approach lacks technical contributions. The reconstruction results are not sufficiently evaluated.

It was our goal to identify the most promising approach to both shape estimation and bone identity representation. The Triplet network approach fit our criteria, both by design and by its success on very large datasets of highly similar objects, although in a different domain. This is why we conducted and extensively evaluated a series of experiments to confirm to which extent

it was possible to apply this approach to bone data.

We evaluated our shape estimation results quantitatively by computing the RMS and Hausdorff distances between the predicted and true 3D shapes, by counting the number of bones in our dataset that would have been better matches than the predicted ones, by comparing the geometric distances between predicted and true 3D shapes with the distances found in literature that uses competing approaches, by applying one such competing approach (SSM) to our data, as well as qualitatively by aligning the predicted and true 3D shapes.

REVIEWERS' COMMENTS:

Reviewer #1 (Remarks to the Author):

I have gone through the response to my comments and I am satisfied with the revised version of the paper.